# Ten-Year Observational Study of Patients with Lung Adenocarcinoma: Clinical Outcomes, Prognostic Factors, and Five-Year Survival Rates

**DOI:** 10.3390/jcm14082552

**Published:** 2025-04-08

**Authors:** Paweł Ziora, Hanna Skiba, Paweł Kiczmer, Natalia Zaboklicka, Julia Wypyszyńska, Maria Stachura, Zuzanna Sito, Mateusz Rydel, Damian Czyżewski, Bogna Drozdzowska

**Affiliations:** 1Department of Pathomorphology, Faculty of Medical Sciences in Zabrze, Medical University of Silesia, 40-055 Katowice, Poland; hskiba@sum.edu.pl; 2Private Medical Practice Paweł Kiczmer, 41-705 Ruda Śląska, Poland; pawel.kiczmer@protonmail.com; 3Students’ Scientific Society, Department of Pathomorphology, Faculty of Medical Sciences in Zabrze, Medical University of Silesia, 40-055 Katowice, Poland; nalciazaboklicka@gmail.com (N.Z.); jul.ale.wy@gmail.com (J.W.); mariastachuraaa@gmail.com (M.S.); zuzia.sito@gmail.com (Z.S.); 4Department of Thoracic Surgery, Faculty of Medical Sciences in Zabrze, Medical University of Silesia, 40-055 Katowice, Poland; mrydel@sum.edu.pl (M.R.); dczyzewski@sum.edu.pl (D.C.)

**Keywords:** lung adenocarcinoma, lung adenocarcinoma survival

## Abstract

**Background/Objectives:** Lung carcinoma is the leading cause of cancer-related deaths globally, with lung adenocarcinoma being the most prevalent subtype. This study aims to review the clinical data and survival outcomes of patients diagnosed with lung adenocarcinoma who underwent surgical treatment. **Methods:** We retrospectively analyzed 471 patients (mean age 65.9 ± 7.81 years, range 38–86; 53.5% women) with histopathologically confirmed lung adenocarcinoma who underwent a lobectomy, bilobectomy, or pneumonectomy between May 2012 and December 2022. All patients were followed for up to five years post-surgery. Their medical histories, including previous neoplasms, comorbidities, tumor characteristics, and symptoms, were thoroughly reviewed. We calculated the overall survival rate and evaluated the impact of tumor grading and spread through air spaces (STAS) on patient outcomes. **Results:** The survival rate for the entire cohort was 76.23%. No significant survival differences emerged between G1 and G2 tumors, whereas both showed markedly better survival rates than G3 tumors. When these findings were applied to a simplified two-tier grading system (low grade vs. high grade), survival analyses showed a clear stratification of prognosis. Patients with STAS had a lower survival rate than those without STAS. **Conclusions:** Our findings indicate that a simplified grading system may improve prognostic evaluations for lung adenocarcinoma patients. Furthermore, STAS is a crucial factor affecting survival rates and should be considered in future treatment strategies. Expanding research in this area is essential to enhance treatment approaches for lung adenocarcinoma patients.

## 1. Introduction

Lung carcinoma remains one of the most prevalent and lethal forms of carcinoma worldwide, accounting for a significant number of cancer-related deaths each year [1,2]. According to GLOBOCAN 2020 statistics, there were 1.8 million reported deaths and 2.2 million new cases in 2020 [2]. Generally, lung carcinoma can be categorized into small-cell lung carcinoma and non-small-cell lung carcinoma (NSCLC), with the latter accounting for over 80% of cases [3,4,5]. Within NSCLC, the following main histological types can be distinguished: squamous carcinoma, adenocarcinoma, and large-cell carcinoma [6].

The primary risk factor remains exposure to tobacco smoke, which has been linked to as many as 80–90% of lung carcinoma cases [7,8,9]. Additionally, exposure to arsenates, nitrosamines, and asbestos is associated with lung carcinoma etiology [7]. Genetic factors and air pollution can also contribute to the development of lung carcinoma [7].

Due to the nonspecific nature of early clinical symptoms, over 60% of NSCLC patients are already in the middle or advanced stages of the disease at the time of diagnosis [4,8,10].

Among its various subtypes, adenocarcinoma of the lung has emerged as the most common histological type, representing a distinct challenge in terms of diagnosis, treatment, and patient care [11]. Historically, advanced-stage lung cancer has had a low 5-year survival rate of approximately 15% [4,6].

Adenocarcinoma of the lung itself represents a heterogeneous group with diverse histopathological features. While tumor grading is widely used to predict outcomes, existing three-tier systems often fail to distinguish meaningful survival differences between G1 and G2 categories. Moreover, the recognition of spread through air spaces (STAS) in recent years has reshaped our understanding of locoregional spread and its effects on patient prognosis. Despite these advances, relatively few large, single-institution cohorts have integrated both grading refinements and STAS assessment into a unified analysis of long-term survival.

Our study fills this gap by examining ten years of experience with surgically treated lung adenocarcinoma, proposing a two-tier grading scheme, and evaluating STAS as a significant pathologic feature. In doing so, we aim to provide clinicians and researchers with practical insights that could streamline grading, inform postoperative monitoring, and ultimately enhance individualized treatment strategies. We focus exclusively on surgically resected adenocarcinoma patients; non-surgical cases are outside this study’s scope.

## 2. Materials and Methods

### 2.1. Study Design

A study was conducted on a cohort of 471 patients with adenocarcinoma of the lung out of a total of 1051 patients who underwent the radical anatomical resection of lung tissue (lobectomy, bilobectomy, or pneumonectomy) due to lung carcinoma between May 2012 and December 2022. A detailed analysis of medical records for all patients operated on for lung carcinoma at our center was performed and collected in a dedicated database. This database included information on prior medical history, exposure to environmental hazards and stimulants, family cancer history, precise cancer stage assessment, exact lung cancer type diagnosis determined through postoperative, histopathological examination, and perioperative care results. Additionally, each patient who underwent surgery was subsequently monitored through our outpatient clinic, enabling the evaluation of long-term treatment outcomes. The day of the surgical procedure marked the starting point of the observation period, which extended for up to five years post-surgery. Survival data for patients were gathered until 1 May 2022, with all subsequent outcomes considered incomplete.

The inclusion criteria were as follows: histopathologically confirmed primary adenocarcinoma. Only patients who underwent radical anatomical resection (lobectomy, bilobectomy, or pneumonectomy) for primary lung adenocarcinoma were included.

The exclusion criteria were as follows: histopathologically confirmed carcinomas other than adenocarcinoma, secondary lung neoplasms confirmed histopathologically, specific types of adenocarcinomas (colloid-/fetal-/enteric-type carcinomas), and the occurrence of more than one histologically distinct tumor in postoperative material. Non-surgical cases were excluded from this study.

In this study, we defined “smoker” to include both current and former smokers and “non-smoker” to include individuals with no smoking history. Further details regarding the group are presented in the results section. The study received approval from the Bioethics Committee of the Medical University of Silesia in Katowice.

### 2.2. Statistical Analysis

The data were presented as the number of cases with percentage values for categorical variables and the mean ± SD for quantitative variables. The normality assumption of each quantitative variable was evaluated through the graphical interpretation of Q–Q plots and histograms. The Kaplan–Meier method was used to determine the survival probability among groups. In instances where comparisons encompassed more than two groups, the Log-rank test with Mantel correction was utilized. To assess the influence of more than one variable on patients’ survival, the Cox proportional hazard model was performed. *p*-values lower than 0.05 were considered significant. Analysis was performed using the R language (version 4.3.1) in RStudio software (version 4.4.2).

## 3. Results

### 3.1. General Characteristics of the Study Group

In total, 471 patients with histopathologically confirmed primary lung adenocarcinoma formed the study cohort, all of whom were Caucasian Polish individuals. The mean age of the participants was 65.9 years (SD ± 7.81), with ages ranging from 38 to 86. Women composed a slightly higher proportion of the study group than men (53.50% vs. 46.50%, respectively), suggesting a relatively balanced distribution between sexes. Regarding smoking status, most participants (75.58%) were active or former smokers, while 24.42% reported no tobacco use. These findings reflect the well-established link between smoking and lung cancer development while also highlighting that approximately one-quarter of patients developed lung adenocarcinoma without any history of tobacco exposure (Table 1).

### 3.2. History of Previous Neoplastic Diseases

Among the 471 patients, 15.89% had a documented history of malignancies prior to their diagnosis of lung adenocarcinoma (Table 2). The most frequently reported previous tumors were breast, prostate, and urinary bladder cancers. Notably, seven patients (1.48%) had a history of lung cancer, indicating that a small subset of individuals had experienced another lung malignancy in the past. These findings underscore the importance of thorough oncological history-taking, as previous malignancies may influence surveillance strategies, treatment decisions, and long-term follow-up.

### 3.3. Comorbidities Among Patients

Cardiovascular diseases were the most prominent comorbidities within the study group (Table 3). More than half of the patients (54.99%) reported hypertension, whereas coronary artery disease was identified in 18.47%, and a prior myocardial infarction was noted in 7.86%. Non-insulin-dependent diabetes mellitus (NIDDM) emerged as a noteworthy metabolic disorder, affecting 16.99% of the cohort. Meanwhile, conditions such as chronic obstructive pulmonary disease (COPD) were present in 15.50% of the patients, reflecting the interplay between lung cancer and other respiratory conditions that share common risk factors (e.g., smoking, air pollution). Although less frequent, additional comorbidities included heart failure, chronic venous disease, and renal failure, each affecting a small percentage of the cohort. These findings illustrate the clinical complexity of lung adenocarcinoma patients, emphasizing the need for multidisciplinary management.

### 3.4. Symptoms of Lung Adenocarcinoma

Over half (60.30%) of the study participants presented with at least one symptom attributable to lung adenocarcinoma at the time of diagnosis (Table 4). Coughing emerged as the single most prevalent symptom (28.24%). The following additional symptoms were reported less frequently: pain was noted in 5.73% of patients, hemoptysis in 4.46% of patients, and dyspnea in 1.06% of patients. This distribution of symptoms aligns with the existing literature indicating that lung cancer often manifests through nonspecific or subtle clinical signs, which can delay diagnosis and potentially affect prognosis.

### 3.5. Tumor Characteristics in the Study Group

Most patients (84.71%) had no regional lymph node involvement (pN0), and importantly, none had clinically detected distant metastases (pM0) (Table 5). Approximately half of the tumors were relatively small at the time of resection (pT1, 48.62%), whereas 34.6% were classified as pT2. A smaller subset of patients displayed more advanced primary tumors (pT3: 10.83%; pT4: 5.94%), indicating more extensive local growth. In keeping with the low rate of nodal disease and the absence of distant metastases in our cohort, early-stage disease (I or II) predominated, with just over 88% of patients presenting at stages I or II. This high proportion of early-stage diagnoses is attributable in part to the fact that every patient included in this analysis underwent surgical resection.

In terms of tumor grading, the largest subset (47.77%) was classified as G3, whereas G2 constituted 44.80% of cases and G1 only 7.43% of cases.

Regarding specific histological patterns, acinar was identified in the majority of cases (77.33%), followed by solid (46.22%), lepidic (28%), micropapillary (24%), and papillary (18%) tumors. Lymphatic invasion was present in 18.68% of patients, and vascular invasion was noted in 15.07%. Pleural invasion was identified in 29.31% of patients overall, with PL1 being the most prevalent category (19.75%). A radical (R0) resection was achieved in 93.21% of the cases, whereas microscopic residual disease (R1) was found in a small fraction (3.82%). Notably, spread through air spaces (STAS) was observed in 22.93% of cases.

### 3.6. Survival Rates

The overall survival analysis is presented in Table 6 and Figure 1.

The overall 5-year survival rate for the entire study cohort was 76.23% (Table 6, Figure 1). One-year survival was 91.9%, declining to 84.3% at the two-year mark. These relatively favorable rates reflect the predominance of early-stage diagnoses and the fact that all patients underwent surgical treatment.

When comparing survival based on tumor grading, there was no significant difference between G1 and G2 survival. Both, however, differed significantly from G3 tumors, which showed considerably poorer survival rates. This prompted the creation of a two-tier grading system—combining G1 and G2 into a single low-grade category, with G3 designated as high grade. The survival analyses demonstrated clear stratification between these two groups (Table 7, Figure 2).

### 3.7. Survival Rates with STAS (Spread Through Air Spaces)

An additional factor influencing prognosis was the presence of spread through air spaces (STAS) (Table 8, Figure 3). Patients whose tumors displayed STAS had worse survival compared to those without this feature. Specifically, after five years, survival for STAS-positive patients was 54.8%, while it was 84.1% for STAS-negative patients.

## 4. Discussion

### 4.1. Symptoms

In our cohort, more than half of the patients presented with at least one nonspecific symptom—most commonly a cough—at the time of diagnosis. Although lung adenocarcinoma can be asymptomatic in its early stages, the appearance of any symptom often prompts earlier diagnostic workup and surgical intervention [12]. According to the WHO, the most common symptoms are persistent coughing, chest pain, shortness of breath, hemoptysis, fatigue, unexplained weight loss, and recurrent lung infections [9].

In our study, coughing was the most common symptom of lung adenocarcinoma, occurring in 28.24% of patients. The remaining manifestations were pain (5.73%), hemoptysis (4.46%), and dyspnea (1.06%). The symptoms observed in our patient group are consistent with other studies [13,14,15].

According to Collins et al., primary tumor manifestations include chest discomfort, coughing, dyspnea, and hemoptysis, aligning with our findings. Our cohort consisted of patients eligible for surgery, thus lacking symptoms linked to tumor metastasis. This factor contributes to the absence of symptoms associated with tumor spread in our group [12].

### 4.2. Staging

The clinical staging of NSCLC is an important factor that influences the choice of the most favorable treatment [16]. We revealed in our study that our patients were most commonly diagnosed with an early stage of lung adenocarcinoma. This group consisted of 416 patients (88.32%), of which 305 (64.75%) had stage I (*n* = 204 IA; *n* = 101 IB), and 111 (23.57%) had stage II carcinoma (*n* = 38 IIA; *n* = 73 IIB). Similar results were also presented in a study by Fassi et al., in which an early stage of lung adenocarcinoma was diagnosed in 61% of patients, as well as in an article by Nasralla et al., which noted that 79.2% of NSCLC patients were diagnosed at stage I and 7.8% at stage II [16,17].

### 4.3. Grading

In line with the latest WHO tumor classification (5th edition), grade 1 tumors comprise lepidic-predominant tumors displaying less than 20% of high-grade patterns. Acinar- or papillary-predominant tumors featuring less than 20% high-grade patterns are designated as G2. Any tumor manifesting over 20% high-grade patterns (solid or micropapillary) falls under the G3 classification [18]. In our study, we found no differences in overall survival between G1 and G2 tumor patients, while G3 tumors were characterized by significantly worse survival compared to the aforementioned cases with G1 and G2. The two-tier system offers a simpler and more straightforward classification scheme, aiding clinicians in ensuring consistent interpretation and application. Furthermore, the two-tier system can reduce interobserver variability and improve interinstitutional agreement. By reducing the number of categories, discrepancies in grading across pathologists are minimized. Consequently, we advocate for the simplification of the grading system. We found that segregating tumors based solely on the presence of 20% or more high-grade patterns enables the division of patients into two groups with notably distinct survival outcomes. Both grading systems retained significance even after adjusting for the stage variable. Moreira et al. suggested a three-tier grading system based on histologic patterns that was found to be a strong prognostic classifier [19].

### 4.4. Vascular and Pleural Invasion

Pleural and vascular invasion constitute the primary pathways through which lung adenocarcinoma spreads [20]. Pleural invasion stands out as an adverse prognostic factor among patients with lung adenocarcinoma. This phenomenon can be classified into the following distinct groups: PL0 (absence of pleural invasion), PL1 (invasion extending beyond the elastic layer of the visceral pleura but without exposure on the pleural surface), PL2 (tumor invasion of the pleural surface), and PL3 (tumor invasion of the parietal pleura) [21].

In our research, pleural invasion was present in 29.31% of cases (PL1 = 19.75%, PL2 = 7.86%, PL3 = 1.7%). Vascular invasion was observed in 15.07% of patients. Ito et al. reported percentages of pleural and vascular invasion at 9% and 12.7%, respectively [22]. Our findings diverge from the study by Usui et al., which indicated a vascular invasion rate of 45.1% [23]. In a study focused on stage I LUAD, vascular invasion was 26% [24]. Poleri et al. reported that 21% of patients exhibited visceral pleura invasion, and 17% showed vascular invasion; however, their study included adenocarcinoma among other NSCLC types [25].

### 4.5. Spread Through Air Spaces (STAS)

STAS is defined as the spread of tumor cells into the air spaces of the lung parenchyma beyond the tumor’s boundary. This phenomenon was initially defined by the WHO in 2015 as a distinctive form of lung adenocarcinoma dissemination [26]. In our study, STAS was detected in 22.93% of patients. Similar results were presented in other studies. According to Huang et al., STAS was present in 28.2–37.3% of cases across all stages of lung adenocarcinoma [27]. Furthermore, Yin et al.’s research indicates that STAS can be found in 14.8 to 56.4% of lung adenocarcinomas [28]. Similar results were also presented in the work by Gu et al., indicating a prevalence of STAS in 15% to 50% of patients [20].

In both of these studies, STAS was proven to be associated with a worse survival rate compared to the group without this route of dissemination, which is consistent with our research. In our study, the difference between the survival rates of patients with STAS and those without it was 86.1% to 89.9% for 1-year survival, 68.8% to 87.0% for 2-year survival, and 54.8% to 85.1% for 5-year survival.

### 4.6. Histological Type

To properly grade and describe lung adenocarcinoma, one must always search for different histological subtypes, as there is usually more than one present. In our study, we found that the most common histological type of lung adenocarcinoma among our patients was acinar (77.33%). The remaining patterns were solid (46.22%), lepidic (28%), micropapillary (24%), and papillary (18%). The percentage of histological subtypes of adenocarcinoma shown in our study differed from those presented in other research; however, some studies overlap with our results, noting that acinar is the most common histologic type [29,30,31]. According to research by Bertoglio et al., lepidic was the most common type, accounting for 43%, while acinar comprised only 11.9% [32].

### 4.7. Survival 

There are many factors affecting survival rates among patients with lung adenocarcinoma. Researchers emphasize the importance of age at diagnosis, sex, TNM staging, tumor size, histological patterns, and factors related to treatment in determining survival rates [33,34]. Garinet et al. claim that staging is one of the most important factors in survival rates [35]. American research calculated the 5-year survival rate for stage IA tumors as 71% and stage IB tumors as 57.6% [33]. Conversely, a study from Turkey demonstrated survival rates of 81.2%, 61.2%, 42.3%, 46.9%, and 37.1% for patients at stages IA, IB, IIA, IIB, and IIIA, respectively [36]. Nevertheless, the 5-year survival rate for our patients did not differentiate between individual tumor stages and was calculated at 76.2%. It is noteworthy that all our patients were surgical candidates, initially not exceeding stage IIIA, and the majority underwent radical (R0) surgeries, which could explain the relatively high survival outcomes. Similar results were presented in the Garinet et al. study, which reported a 5-year survival rate of 70% [35]. Yang et al. suggested a 5-year survival rate of 45-65% [34]. Conversely, Urer et al. reported an inferior 5-year survival rate of 52.3% [36].

### 4.8. Comorbidities

The most commonly reported comorbidities among our patients affected the cardiovascular system, with hypertension being the leading cause, affecting 54.99% of patients. Other cardiovascular diseases included coronary disease (18.47%), past myocardial infarction (7.86%), heart failure, chronic venous disease, and stroke. Similarly to other studies on lung carcinoma, cardiovascular diseases have also emerged as frequent comorbidities, with incidences ranging from 12.9% to 43% [37]. Analogous to lung carcinoma, the risk of developing most CVDs also increases with age [37]. It is noteworthy that approximately 60% of the population develops hypertension by the age of 60, and the mean age within our population was 65.9 years [38].

The second most prevalent group of comorbidities involved the respiratory system, with COPD standing out as the most frequently reported disease, aligning with existing data [37,39,40]. Both lung carcinoma and COPD share similar risk factors, such as tobacco smoke exposure, air pollution, and older age. It is suggested that these two diseases are also closely linked at a molecular level, involving oxidative stress and inflammation [37,41].

Approximately 18% of patients from our database suffered from either insulin-dependent or non-insulin-dependent diabetes mellitus, which is comparable to the results of other studies [37,40,42].

Within our patient cohort, 15.9% reported a history of prior neoplastic diseases, with breast carcinoma being the most frequent, followed by prostate, urinary bladder, uterine, and other lung carcinomas. The prevalence of previous neoplasms corresponds with existing studies on lung carcinoma patients [40,43].

### 4.9. Study Limitations and Clinical Implications

This study has several important limitations. First, it is a retrospective, single-institution analysis focusing exclusively on patients who underwent surgical resection for lung adenocarcinoma. As a result, this cohort comprises predominantly early-stage disease, potentially underrepresenting cases with more advanced or metastatic tumors. The choice to include only surgically treated patients—without comprehensive reporting on systemic therapies or radiation—limits our ability to comment on outcomes related to nonsurgical treatments or combined treatment modalities. In particular, we did not collect detailed information regarding adjuvant or neoadjuvant interventions (e.g., chemotherapy, targeted therapies, immunotherapy, or radiation), nor did we stratify results based on specific treatment approaches. This omission may mask differences in survival outcomes across subgroups, especially when considering the fact that treatment decisions, such as the use of adjuvant-targeted or immunotherapy, can vary widely by patient age, stage, and molecular profile.

Second, data on prognostic biomarkers such as PD-L1 expression or genomic alterations (e.g., EGFR, ALK, KRAS) were unavailable for much of the study period, reflecting the evolving standards of molecular testing over the decade in question. Consequently, we could not account for the influence of precision medicine approaches, which increasingly drive therapeutic strategies and can significantly alter survival outcomes. The absence of these variables further complicates the interpretation of our results, especially given that targeted and immunotherapeutic agents play an expanding role in early-stage, resectable non-small cell lung cancer (NSCLC).

Third, recurrence rates and patterns were not systematically recorded in our database and are, thus, not presented. Recurrence is a critical endpoint that can inform the need for adjuvant therapy or more intensive follow-up, particularly in high-risk subgroups. Omitting these data may limit the applicability of our findings in clinical practice, where understanding the risk of recurrence is crucial for post-surgical management decisions.

Nevertheless, despite these limitations, our publication offers valuable insights into long-term clinical outcomes and relevant tumor characteristics in surgically treated lung adenocarcinoma. By combining a robust ten-year retrospective dataset with detailed histopathological analyses, we provide a substantial evidence base for clinicians and researchers alike and hope that this work will inform future studies seeking to integrate both traditional pathology-based approaches and contemporary precision medicine strategies.

## 5. Conclusions

The symptoms of lung cancer were evident in over 60% of patients, with a cough being the most frequently reported symptom at 28.24%. Notably, 88.32% of our patients were diagnosed at an early stage of lung adenocarcinoma. It was found that the presence of STAS (spread through air spaces) is associated with a worse survival rate compared to those without this route of spread. Recognizing the presence of STAS may inform postoperative monitoring and the potential need for adjuvant therapy. Among our patients, the acinar histological type of lung adenocarcinoma was the most common characteristic, accounting for 77.33%. The overall survival rate for patients with adenocarcinoma undergoing surgery was 76.23%. The lack of significant survival differences between G1 and G2 patients suggests that a two-tier grading system might be more appropriate than the current three-tier system. In light of these findings, we emphasize the importance of validating our two-tier approach in larger, prospective, multi-institutional studies—particularly those integrating comprehensive molecular biomarker data and standardized adjuvant treatments. Additionally, the most prevalent comorbidities reported predominantly affected the cardiovascular system, followed by the respiratory system, with COPD being the most common respiratory comorbidity.

## Figures and Tables

**Figure 1 jcm-14-02552-f001:**
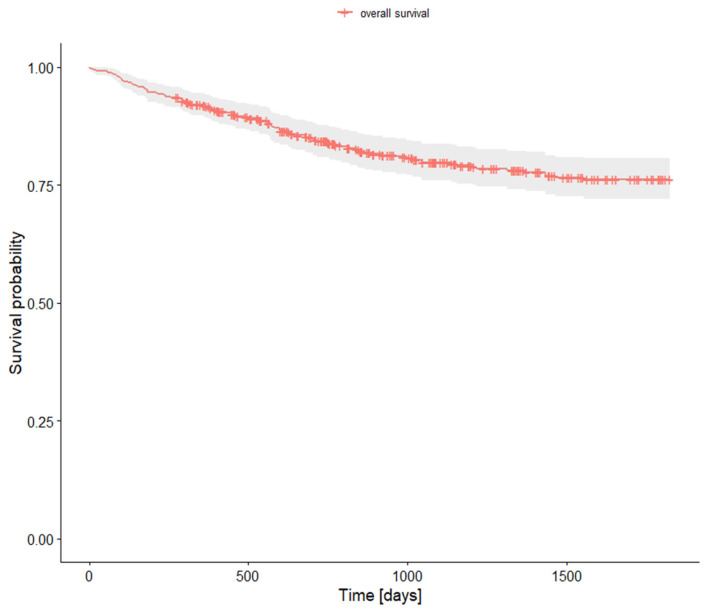
Survival analysis.

**Figure 2 jcm-14-02552-f002:**
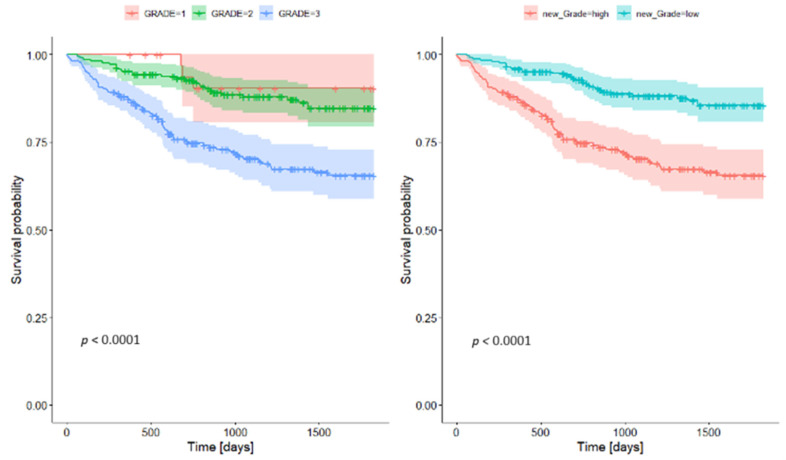
Survival rates for grading.

**Figure 3 jcm-14-02552-f003:**
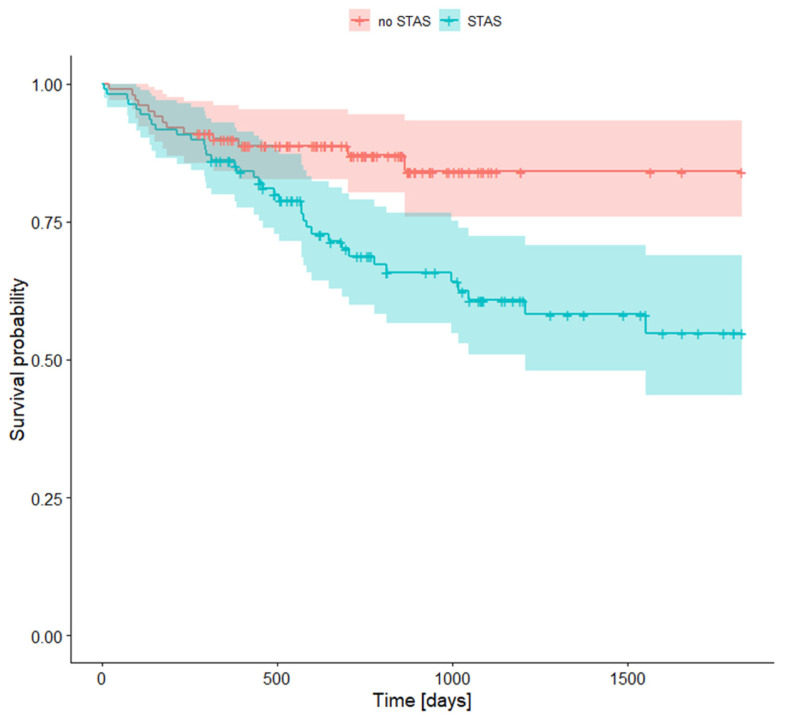
Survival rates for STAS presence.

**Table 1 jcm-14-02552-t001:** General characteristics of the study group.

	Mean	SD	95% CI	95% CI
Age	65.9	7.81	65.19	66.61
Gender	*n*	%	95% CI	95% CI
Female	252.00	53.50%	49.00%	58.01%
Male	219.00	46.50%	41.99%	51.00%
Smoking Status	*n*	%	95% CI	95% CI
Smoker	356	75.58%	71.70%	79.46%
Non-smoker	115	24.42%	20.54%	28.30%

**Table 2 jcm-14-02552-t002:** Former cancer history among patients.

Former Neoplastic Diseases	*n*	%	95% CI	95% CI
Any former neoplastic diseases	75	15.89%	12.59%	19.19%
Breast	16	3.39%	1.76%	5.02%
Prostate	12	2.54%	1.12%	3.96%
Urinary bladder	10	2.12%	0.82%	3.42%
Uterus	7	1.48%	0.39%	2.57%
Lung	7	1.48%	0.39%	2.57%
Large intestine	5	1.06%	0.13%	1.98%
Kidney	4	0.85%	0.02%	1.68%
Larynx	2	0.42%	0.00%	1.01%
Ovary	2	0.42%	0.00%	1.01%
Stomach	2	0.42%	0.00%	1.01%
Other	12	2.54%	1.12%	3.96%
Neoplasms in family	102	21.61%	17.89%	25.33%

**Table 3 jcm-14-02552-t003:** Comorbidities among patients.

Comorbidities	*n*	%	95% CI	95% CI
Insulin-dependent diabetes mellitus, IDDM	4.00	0.85%	0.02%	1.68%
Non-insulin-dependent diabetes mellitus, NIDDM	80.00	16.99%	13.59%	20.38%
Myocardial infarction in the past	37.00	7.86%	5.43%	10.29%
Heart failure	7.00	1.49%	0.39%	2.58%
Renal failure	3.00	0.64%	0.00%	1.36%
COPD	73.00	15.50%	12.23%	18.77%
Bronchial asthma	22.00	4.67%	2.77%	6.58%
Epilepsy	0.00	0.00%	0.00%	0.00%
Stroke in the past	1.00	0.21%	0.00%	0.63%
Hypertension	259.00	54.99%	50.50%	59.48%
Coronary disease	87.00	18.47%	14.97%	21.98%
Blood coagulation disorders	1.00	0.21%	−0.20%	0.63%
Chronic venous disease	7.00	1.49%	0.39%	2.58%

**Table 4 jcm-14-02552-t004:** Cancer symptoms among patients.

Cancer Symptoms	*n*	%	95% CI	95% CI
Any cancer symptoms	284.00	60.30%	55.88%	64.72%
Cough	133	28.24%	24.17%	32.30%
Pain	27.00	5.73%	3.63%	7.83%
Hemoptysis	21.00	4.46%	2.59%	6.32%
Dyspnea	5.00	1.06%	0.14%	1.99%

**Table 5 jcm-14-02552-t005:** Tumor characteristics.

	*n*	%	95% CI	95% CI
**pT Parameter**				
pT1a	17	3.61%	1.92%	5.29%
pT1b	107	22.72%	18.93%	26.50%
pT1c	105	22.29%	18.53%	26.05%
pT2a	118	25.05%	21.14%	28.97%
pT2b	45	9.55%	6.90%	12.21%
pT3	51	10.83%	8.02%	13.63%
pT4	28	5.94%	3.81%	8.08%
**pN Parameter**				
pN0	399	84.71%	81.46%	87.96%
pN1	41	8.70%	6.16%	11.25%
pN2	31	6.58%	4.34%	8.82%
**pM Parameter**				
pM0	471	100.00%	100.00%	100.00%
**Stage**				
IA1	17	3.61%	1.92%	5.29%
IA2	97	20.59%	16.94%	24.25%
IA3	90	19.11%	15.56%	22.66%
IB	101	21.44%	17.74%	25.15%
IIA	38	8.07%	5.61%	10.53%
IIB	73	15.50%	12.23%	18.77%
IIIA	38	8.07%	5.61%	10.53%
IIIB	16	3.40%	1.76%	5.03%
IVA	1	0.21%	0.00%	0.63%
**Grading**				
G1	35	7.43%	5.06%	9.80%
G2	211	44.80%	40.31%	49.29%
G3	225	47.77%	43.26%	52.28%
**Vascular invasion**				
Lymphatic invasion	88	18.68%	15.16%	22.20%
Vascular invasion	71	15.07%	11.84%	18.31%
**Pleural invasion**				
0	330	70.06%	65.93%	74.20%
1	93	19.75%	16.15%	23.34%
2	37	7.86%	5.43%	10.29%
3	8	1.70%	0.53%	2.87%
**Radical resection**				
R0	439	93.21%	90.93%	95.48%
R1	18	3.82%	2.09%	5.55%
**Spread through air spaces**				
STAS	108	22.93%	19.13%	26.73%
**Histologic pattern**				
	*n*	%	95% CI	95% CI
Lepidic	126	28%	23.85%	32.15%
Acinar	348	77.33%	73.46%	81.20%
Papillary	81	18%	14.45%	21.55%
Micropapillary	108	24%	20.05%	27.95%
Solid	208	46.22%	41.61%	50.83%

**Table 6 jcm-14-02552-t006:** Survival analysis.

	Research Population		
	Survival Rate	95% CI	
**1-year survival rate**	0.919000	0.895	0.944
**2-year survival rate**	0.843277	0.81	0.878
**5-year survival rate**	0.762390	0.721	0.806

**Table 7 jcm-14-02552-t007:** Survival rates for grading.

	Low Grade			High Grade		
	Survival Rate	95% CI		Survival Rate	95% CI	
**1-year survival rate**	0.959	0.935	0.984	0.875	0.833	0.919
**2-year survival rate**	0.928	0.896	0.962	0.747	0.69	0.809
**5-year survival rate**	0.865	0.808	0.906	0.655	0.588	0.729

**Table 8 jcm-14-02552-t008:** Survival rates for STAS presence.

	STAS +			STAS −		
	Survival Rate	95% CI		Survival Rate	95% CI	
**1-year survival rate**	0.861	0.798	0.929	0.899	0.842	0.961
**2-year survival rate**	0.688	0.599	0.790	0.870	0.803	0.943
**5-year survival rate**	0.548	0.436	0.688	0.841	0.758	0.934

## Data Availability

The data presented in this study are available in this article.

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
