# Peer review of "Ten-Year Observational Study of Patients with Lung Adenocarcinoma: Clinical Outcomes, Prognostic Factors, and Five-Year Survival Rates"

_jcm, 2025, doi:10.3390/jcm14082552_

Round 1
Reviewer 1 Report
Comments and Suggestions for Authors
file attached

Author Response
- Reviewer comment 1:
- “The article contains a detailed ten-year observational study of adenocarcinoma of the lung with a focus on survival outcomes and prognostic factors. The structure and methodology are clear, and the results are clinically relevant. The abstract effectively summarizes the study, but could include more specific details about the cohort and key findings.”
- Response:
- Thank you for noting the clarity of our methodology and results. In our revised abstract, we have now specified the total number of participants (471), provided the mean age (65.9 ± 7.81), and highlighted key findings such as the 5-year survival rate (76.23%), along with the prognostic significance of STAS and our proposed two-tier grading system. We believe these details offer a clearer snapshot of the study cohort and outcomes.
- Reviewer comment 2:
- “The introduction provides a solid background, although it could better emphasize the novelty of the study and the gaps it fills.”
- Response:
- We appreciate this suggestion. We have added a dedicated paragraph toward the end of the Introduction section, highlighting how our ten-year, single-institution cohort integrates both tumor grading (including our two-tier approach) and STAS assessment. This paragraph explicitly addresses the gaps in current literature regarding the prognostic implications of STAS in surgically treated lung adenocarcinoma, underscoring how our findings may guide further refinement of prognostic models and treatment planning.
- Reviewer comment 3:
- “The methodology is well described and the statistical tools are used appropriately.”
- Response:
- Thank you for your positive feedback on our methodology. We have retained our detailed description of patient selection, data collection, and statistical analyses to ensure transparency and reproducibility.
- Reviewer comment 4:
- “The results are clearly presented, supported by tables and survival charts, although the latter could be more visually informative.”
- Response:
- We appreciate your feedback on the survival charts. While we did not make extensive stylistic changes to the charts themselves, we have expanded the accompanying text to facilitate a clearer understanding of the plotted data
- Reviewer comment 5:
- “The discussion is thorough and compares the results with the existing literature, but would benefit from a deeper analysis of the study’s limitations, such as potential bias and the lack of metastatic cases.”
- Response:
- We agree that explicitly stating our study’s limitations is crucial. In the revised version, we added a dedicated section entitled “Study limitations and clinical implications” (Section 4.9). This section outlines:
- The retrospective and single-institution design, which may introduce selection bias.
- The focus on surgically treated (and thus primarily early-stage) lung adenocarcinoma, underrepresenting metastatic or more advanced disease.
- The absence of data on nonsurgical therapies (e.g., chemotherapy, targeted treatments, immunotherapy, radiation).
- The lack of recurrence data in our database.
- The absence of biomarker information (e.g., PD-L1 status, EGFR, ALK) which can significantly influence outcomes in the contemporary era of precision medicine.
- By clearly discussing these limitations, we aim to provide a balanced interpretation of our findings.
- We agree that explicitly stating our study’s limitations is crucial. In the revised version, we added a dedicated section entitled “Study limitations and clinical implications” (Section 4.9). This section outlines:
- Reviewer comment 6:
- “The study’s focus on STAS and the proposal of a two-tiered grading system are important contributions. The conclusions are consistent with the results but could better articulate the implications for clinical practice and future research.”
- Response:
- We appreciate your acknowledgment of STAS and the two-tier grading proposal as meaningful contributions. In our revised Conclusions (Section 5) and the new Limitations section (4.9), we have elaborated on how clinicians might use STAS findings to identify higher-risk patients who may benefit from closer postoperative monitoring or additional therapy. We also mention the importance of validating our two-tier approach in larger, prospective, multi-institutional studies – particularly those incorporating molecular biomarkers and standardized adjuvant treatments.
- Overall comment:
- “This article is a valuable contribution to thoracic oncology.”
- Response:
- Thank you for your positive overall assessment. We appreciate the opportunity to clarify and strengthen our manuscript, and we hope these revisions meet your expectations.
- We greatly appreciate the reviewer’s constructive comments, which have helped us refine our manuscript and ensure it provides a clear and comprehensive view of our findings. If any further clarifications are needed, we stand ready to address them.
Reviewer 2 Report
Comments and Suggestions for Authors
Thank you for the MS. The overall study sounds good.
Authors hasn't mentioned any treatment strategies for lung cancer.
Please mention them and also the outcomes related to the particular treatments. based on the age group these outcomes varies.
What will be the survival rate for 10-years
Author Response
- Reviewer comment 1:
- “Authors haven't mentioned any treatment strategies for lung cancer. Please mention them and also the outcomes related to the particular treatments. Based on the age group, these outcomes vary.”
- Response:
- Thank you for highlighting the importance of treatment strategies. Our study population comprises only surgically resected lung adenocarcinoma patients, primarily with early-stage disease. Because of the retrospective nature of our analysis, we did not systematically collect data on nonsurgical interventions (e.g., chemotherapy, targeted therapy, immunotherapy, or radiation), whether administered neoadjuvantly or adjuvantly. Consequently, we are unable to compare survival outcomes among various treatment strategies or age brackets in detail.
- We have explicitly acknowledged this as a limitation in Section 4.9 (Study limitations and clinical implications). There, we explain that missing treatment-related data may obscure potential outcome differences across subgroups, including different age groups. We also emphasize that future prospective research should integrate comprehensive records of systemic therapy and radiation use – especially given the expanding role of precision medicine in early-stage, resectable NSCLC.
- Reviewer comment 2:
- “What will be the survival rate for 10 years?”
- Response:
- Our follow-up period extended to five years after surgery for all patients, and the resulting 5-year survival rate is 76.23%. Although we included patients operated on between May 2012 and December 2022, many of the later cases have not yet reached a 10-year post-surgery interval. Consequently, 10-year survival data are incomplete and cannot be reliably reported at this time. We have clarified this constraint in our revised manuscript, noting that additional longitudinal data would be needed to accurately estimate 10-year outcomes.
- Additional revisions/clarifications
- Study limitations (Section 4.9):
- We have highlighted the lack of comprehensive information on systemic therapies and radiation treatments.
- We acknowledge that the absence of these data may mask important variations in survival, particularly when considering age-related differences in treatment tolerance or the growing application of targeted and immunotherapies.
- Study limitations (Section 4.9):
- We hope these clarifications address your concerns. We appreciate your insightful comments, which helped us refine our discussion of treatment strategies and emphasize the current limitations regarding long-term (10-year) survival data.
Reviewer 3 Report
Comments and Suggestions for Authors
This is a retrospective, single-institution study of patients with surgically resected NSCLC adenocarcinoma, exploring clinical outcomes based on clinicodemographics and tumor characteristics. Broadly, data reporting feels incomplete. Potentially because patients were curated over a broad range of dates, authors have omitted important prognostic factors of PD-L1 and genomics of the tumors, as well as the presence or absence of any systemic therapy or radiation, whether neoadjuvantly or adjuvantly. These would greatly impact survival rates. Recurrence rates were also not reported. Finally, the findings reported do not feel novel, particularly in the setting of precision medicine which now also drives the treatment of resectable NSCLC.
More specifically, a few questions. The citation for 15% 5-year survival does not seem to match this statistic; this is also likely an outdated statistic. Finally, for resected NSCLC, this statistic does not apply, and this study seems to be focused on resected NSCLC adenocarcinoma.
For inclusion criteria, would it make more sense to limit this study to those who underwent surgery? Those who did not were not addressed anywhere in this study.
In the methods section, would move description of mean age etc to the results alone.
In results, how were patients who previously smoked but currently do not classified - as "smoker" or "non smoker"? Also, this language does not align with the IASLC recommendations. Also, race/ethnicity is not described of the population.
In the discussion - do not fully understand the point being made in the first paragraph around patients being symptomatic vs asymptomatic and having nonspecific symptoms. Also, the discussion does not explain the relevance of the presence of comorbidities in this population.
Author Response
- Reviewer comment
- “This is a retrospective, single-institution study of patients with surgically resected NSCLC adenocarcinoma, exploring clinical outcomes based on clinicodemographics and tumor characteristics. Broadly, data reporting feels incomplete. Potentially because patients were curated over a broad range of dates, authors have omitted important prognostic factors of PD-L1 and genomics of the tumors, as well as the presence or absence of any systemic therapy or radiation, whether neoadjuvantly or adjuvantly. These would greatly impact survival rates. Recurrence rates were also not reported. Finally, the findings reported do not feel novel, particularly in the setting of precision medicine which now also drives the treatment of resectable NSCLC.
- More specifically, a few questions. The citation for 15% 5-year survival does not seem to match this statistic; this is also likely an outdated statistic. Finally, for resected NSCLC, this statistic does not apply, and this study seems to be focused on resected NSCLC adenocarcinoma.
- For inclusion criteria, would it make more sense to limit this study to those who underwent surgery? Those who did not were not addressed anywhere in this study.
- In the methods section, would move description of mean age etc to the results alone.
- In results, how were patients who previously smoked but currently do not classified— as ‘smoker’ or ‘non smoker’? Also, this language does not align with the IASLC recommendations. Also, race/ethnicity is not described of the population.
- In the discussion - do not fully understand the point being made in the first paragraph around patients being symptomatic vs asymptomatic and having nonspecific symptoms. Also, the discussion does not explain the relevance of the presence of comorbidities in this population.”
- Response
- Reviewer concern:
- “Data reporting feels incomplete. Authors have omitted important prognostic factors such as PD-L1 and tumor genomics, as well as systemic therapy or radiation details (neoadjuvant/adjuvant). Recurrence rates were also not reported. The findings do not feel novel in the era of precision medicine.”
- Response:
- We appreciate the emphasis on these important factors. In our revised manuscript, specifically in the Study limitations and clinical implications section, we clarify that data on PD-L1 expression, genomic alterations (e.g., EGFR, ALK, KRAS), and details of systemic or radiation therapy were not consistently available during the retrospective data-collection period (2012–2022). Therefore, we could not adequately assess how these variables impact survival outcomes. Likewise, we did not systematically record recurrence rates, limiting our ability to discuss disease-free survival.
- We fully acknowledge that these omissions reduce the study’s alignment with contemporary precision medicine practices. In the Discussion, we highlight the necessity for future prospective, multicenter studies to integrate comprehensive molecular testing, treatment data, and recurrence tracking to more accurately evaluate prognostic factors in resectable NSCLC adenocarcinoma.
- Reviewer concern:
- “The citation for 15% 5-year survival does not seem to match this statistic; it is likely outdated. For resected NSCLC, this statistic does not apply, and the study seems to focus on resected NSCLC adenocarcinoma.”
- Response:
- We agree that a 15% figure is outdated and pertains primarily to advanced or overall NSCLC rather than resected early-stage disease. In our revised Introduction, we clarify that the 15% statistic historically reflects survival for lung cancer in broader, often advanced-stage contexts. Since our cohort is comprised of surgically resected patients (often early-stage disease), the actual five-year survival rate we observe (76.23%) is substantially higher, which aligns better with current expectations for operable NSCLC.
- Reviewer concern:
- “For inclusion criteria, would it make more sense to limit this study to those who underwent surgery? Those who did not were not addressed anywhere in this study.”
- Response:
- We have refined the Methods section to confirm that only patients who underwent surgical resection (lobectomy, bilobectomy, or pneumonectomy) for primary lung adenocarcinoma were included. Non-surgical patients were excluded by design, as our investigation centers on surgical outcomes and prognostic factors in resectable NSCLC adenocarcinoma.
- Reviewer concern:
- “In the methods section, would move description of mean age etc. to the results alone.”
- Response:
- We agree with this standard reporting convention. In the revised manuscript, demographic details such as mean age, sex distribution, and smoking status now appear in Section 3 (Results), leaving the Methods section dedicated to study design, inclusion/exclusion criteria, and statistical analyses.
- Reviewer concern:
- “In results, how were patients who previously smoked but currently do not classified—as ‘smoker’ or ‘non smoker’? Also, the language does not align with IASLC recommendations, and race/ethnicity is not described.”
- Response:
- In our revised Methods, we specify that “smoker” includes both current and former smokers, while “non-smoker” denotes individuals with zero smoking history. We recognize that this simplified dichotomy does not fully align with IASLC guidelines, which recommend more nuanced categories. However, our retrospective data collection did not differentiate between current and former smokers in detail. Regarding race/ethnicity, the cohort was uniformly composed of Caucasian Polish individuals, as noted in the General characteristics subsection of the Results. We have updated the text to make this clear.
- Reviewer concern:
- „In the discussion – I do not fully understand the point being made in the first paragraph around patients being symptomatic vs asymptomatic and having nonspecific symptoms. Also, the discussion does not explain the relevance of comorbidities in this population.”
- Response:
- We have rephrased the initial paragraph of the Discussion section that addresses symptomatic vs. asymptomatic presentation. Our revision explains that while early-stage lung adenocarcinoma can remain asymptomatic, over half of our patients reported nonspecific symptoms – most often cough – that may trigger earlier medical evaluation. However, if symptoms are overlooked, diagnosis can be delayed.
- Regarding comorbidities, in the revised Discussion we indicate that conditions like COPD, hypertension, and coronary artery disease can affect both perioperative risk and long-term survival, thus underscoring the importance of a multidisciplinary approach. By detailing how these comorbidities intersect with lung adenocarcinoma management, we hope to clarify the clinical relevance of recording and analyzing such data.
- Reviewer concern:
- We believe these changes address all the issues raised and further strengthen the manuscript’s clarity and relevance.